# Specific Biomarkers in Spinocerebellar Ataxia Type 3: A Systematic Review of Their Potential Uses in Disease Staging and Treatment Assessment

**DOI:** 10.3390/ijms25158074

**Published:** 2024-07-24

**Authors:** Alexandra E. Soto-Piña, Caroline C. Pulido-Alvarado, Jaroslaw Dulski, Zbigniew K. Wszolek, Jonathan J. Magaña

**Affiliations:** 1Facultad de Medicina, Universidad Autónoma del Estado de México, Toluca 50180, Mexico; aesotop@uaemex.mx (A.E.S.-P.); carolpulidoalvarado@gmail.com (C.C.P.-A.); 2Department of Neuroscience, Mayo Clinic, Jacksonville, FL 32224, USA; 3Department of Neurology, Mayo Clinic, Jacksonville, FL 32224, USA; jaroslaw.dulski@gumed.edu.pl; 4Division of Neurological and Psychiatric Nursing, Faculty of Health Sciences, Medical University of Gdansk, 80-211 Gdansk, Poland; 5Neurology Department, St Adalbert Hospital, Copernicus PL Ltd., 80-462 Gdansk, Poland; 6Department of Genomic Medicine, Instituto Nacional de Rehabilitación—Luis Guillermo Ibarra, Ibarra, Ciudad de México 14389, Mexico; jmagana@inr.gob.mx; 7Department of Bioengineering, School of Engineering and Sciences, Tecnológico de Monterrey, Campus Ciudad de México, Ciudad de México 14380, Mexico

**Keywords:** spinocerebellar ataxia type 3, Machado–Joseph disease, biomarkers, neurofilament light chain, PolyQ-ATXN3

## Abstract

Spinocerebellar ataxia type 3 (SCA3) is the most common type of disease related to poly-glutamine (polyQ) repeats. Its hallmark pathology is related to the abnormal accumulation of ataxin 3 with a longer polyQ tract (polyQ-ATXN3). However, there are other mechanisms related to SCA3 progression that require identifying trait and state biomarkers for a more accurate diagnosis and prognosis. Moreover, the identification of potential pharmacodynamic targets and assessment of therapeutic efficacy necessitates valid biomarker profiles. The aim of this review was to identify potential trait and state biomarkers and their potential value in clinical trials. Our results show that, in SCA3, there are different fluid biomarkers involved in neurodegeneration, oxidative stress, metabolism, miRNA and novel genes. However, neurofilament light chain NfL and polyQ-ATXN3 stand out as the most prevalent in body fluids and SCA3 stages. A heterogeneity analysis of NfL revealed that it may be a valuable state biomarker, particularly when measured in plasma. Nonetheless, since it could be a more beneficial approach to tracking SCA3 progression and clinical trial efficacy, it is more convenient to perform a biomarker profile evaluation than to rely on only one.

## 1. Introduction

Spinocerebellar ataxia type 3 (SCA3) or Machado–Joseph Disease (MJD) is the most common form of SCA [1,2]. It is a poly-glutamine (polyQ) disorder with a prevalence of 1–5/100,000 [3] caused by the expansion of an unstable CAG trinucleotide repeat located in the *ATXN3* gene [4]. As a result, an ATXN3 protein with a longer Q tract (polyQ-ATXN3), related to altered protein misfolding, accumulation and polyQ-ATXN3 inclusion formation in neurons, is generated, causing gain-of-function toxicity [5,6,7,8]. This neuropathology mainly affects the cerebellum and other regions that control motor functions [9,10].

The complexity of SCA3 encompasses not only neuropathological features but also a wide range of clinical signs and chemical modifications [11,12]. The complexity of the disease, as well as the demand for a therapeutic strategy to treat it, requires studying trait and state biomarkers, the former portraying features before disease onset and the latter revealing the condition during disease progression or treatment [13]. Other biomarkers besides polyQ-ATXN3 are thus required for comprehending disease progression and the pathways that may aid in management and therapeutic intervention for patients.

From a clinical point of view, most assessments have relied on using medical scales such as the Assessment and Rating of Ataxia (SARA) and the Inventory of Non-Ataxia Symptoms (INAS) scales [14,15,16]. While these can aid the diagnosis of patients with explicit ataxic symptoms, the asymptomatic and preclinical stages require a more comprehensive battery of assessments, hence why the use of fluid markers represents a first option for inclusion in this analysis.

Current clinical trials have implemented the use of SARA and ICARS scores to evaluate the efficacy of some compounds like riluzole, troriluzole, trehalose and valproic acid [14,15,17,18,19], as well as antisense olinucleotides (ASOS) [20]. Nonetheless, these clinical trials have not yet included specific and sensitive biomarkers that can improve disease progression and therapeutic efficacy monitoring. Because of their easy availability in clinical practice, the use of fluid biomarkers can therefore potentially aid in achieving these goals. In this investigation, we present an overview of SCA3 biomarkers measured in the body fluids of patients in different SCA3 stages. The aim of this study was to identify potential trait and state biomarkers for clinical stage classification, prognosis and therapeutic intervention assessment.

## 2. Materials and Methods

### 2.1. Literature Search Strategy

This systematic review was written according to the Preferred Reporting Items for Systematic Reviews and Meta-analyses (PRISMA) guidelines and registered in PROSPERO (ID: CRD42023424952). We performed a search in PUBMED with the terms (“Spinocerebellar ataxia type 3”, “Machado–Joseph Disease”, “SCA3”) AND (“biomarker” OR “biofluid” OR “pharmacodynamic marker”). The manuscript inclusion criteria consisted of cross-sectional and longitudinal studies involving measures of protein, DNA and RNA markers in body fluids such as plasma, serum, urine and cerebrospinal fluid (CSF). A total of 85 manuscripts appeared under these criteria as of 4 July 2024. The record discrimination strategy was performed in Evidence for Policy and Practice reviewer version 6 (EPPI v6). Firstly, manuscripts were screened for duplicates and excluded by title, abstract and full text according to the following criteria: reviews or manuscripts with no quantitative data, tissue measurements, cell or animal models and studies that did not include biofluids (Figure 1). Ultimately, 24 studies complied with the inclusion criteria and the STROBE guidelines for cohorts and case–control studies.

### 2.2. Data Extraction and Management

Data were extracted from eligible studies according to biomarker type, fluid source, outcome, biomarker-related variables and manuscript reference. Data were pooled in tables according to the stage classification proposed by Mass et al. [21]. “Asymptomatic” refers to carriers who are free of symptoms and signs. “Preclinical” refers to carriers with a SARA score < 3 with some early unspecific symptoms (e.g., muscle cramps, pain and fatigue) [22]. In this category, we also included data from “preataxic” patients with non-ataxic signs [23]. Ataxic patients included those who had a SARA score ≥ 3, as well as alterations in balance, gait, coordination, lower limb spasticity, vision, cognition, dysarthria, dysphagia and structural brain abnormalities. Moreover, data from asymptomatic carriers were also considered to show biomarker progression, as shown in Figure 3a,b, specifically for polyQ-ATXN3 and NfL; the rest of the biomarkers were based on preataxic and ataxic stage information. Table 1 and Appendix A show a summary of the studies in which there were significant correlations between biomarkers and clinical variables.

### 2.3. Statistical Analysis of Neurofilament Light Chain (NfL) Levels

For NfL estimations, we included studies that used the same detection assay platform (Single Molecule Array: SIMOA); data from two cohorts were excluded on the grounds that they were validation cohorts in two studies that were already included as main cohorts. We compared NfL between controls and preclinical and ataxic subjects using a random effect analysis for continuous outcome variables [24]. The means of NfL for each separate group, as well as the differences in means between groups, were estimated, along with 95% confidence intervals (CIs). I^2^ is a reliable descriptive statistic for analyzing homogeneity, particularly when accompanied by other statistics like means and *p*-values [25]. The between-study heterogeneity in means and mean differences were examined by estimating the I^2^ statistic, which measures the proportion of variation in these quantities due to heterogeneity beyond chance [26]. *p*-values < 0.05 were considered statistically significant. All the statistical tests were two-sided. Statistical analyses were performed using R Statistical Software (version 4.1.2; R Foundation for Statistical Computing, Vienna, Austria).

## 3. Results and Discussion

Figure 2 summarizes the useful SCA3 fluid biomarkers for delimiting disease stages. However, none of these have been used to evaluate therapeutic efficacy in clinical trials, though some are involved in oxidative stress, metabolism, neurodegeneration and RNA metabolism. There are also biomarkers like miRNAs that researchers have started to explore for tracking disease progression and pharmacodynamic use.

The total number of patients analyzed in this review was 2838; of these, 1320 were ataxic, 168 were preataxic and 1350 were healthy controls. In these groups, the most frequently tested biomarkers were NfL (n = 8/21) and PolyQ-ATXN3 (n = 3/21) (Table 1).

The most widely analyzed marker in SCA3 was NfL (Table 1). As shown in Appendix A, I_2_ was 0% in NfL plasma concentrations from both preclinical and ataxic patients. The values in the serum were higher than 90%; in the CSF, they were 0%, and in control and ataxic patients, they were 53%. This indicates that I_2_ could be a reliable approach to evaluating heterogeneity in plasma and CSF and in comparisons between preclinical/ataxic and control/ataxic patients, respectively. This result is reinforced by the results presented in Appendix A, where the mean difference between preclinical/ataxic and control/ataxic patients in plasma and CSF was statistically significant (*p* < 0.0001). However, an overestimation of 12% is possible if I_2_ is high and there are fewer than seven studies analyzed [27]. Therefore, the difference between plasma and CSF could be due to difficulties in recruiting patients for lumbar puncture. Nonetheless, in these cases, NfL can be used as a stage-discriminative biomarker, something which is consistent with a meta-analysis showing that NfL levels increase in patients with different types of ataxia [28] and cerebellum atrophy disorders [29]. It is thought that axonal damage allows NfL release and protein synthesis and secretion as axons attempt to regenerate [30]. Therefore, NfL in CSF might also be an early disease state biomarker and its elevation could reflect axonal damage in SCA3. While it was not possible to identify changes in NfL levels in CSF through meta-analysis, their discriminative capacity in CSF and plasma has been confirmed via area under the ROC curve [31]. Moreover, as a biomarker, NfL has more positive correlations with clinical variables like SARA score, age and CAG repeats (Table 1) and negative correlations with pons, midbrain and brainstem volumes [32]. Further longitudinal NfL studies should be performed to better understand the patterns in the dynamics of this biomarker throughout patient lifespans in distinct populations, as can be seen in other polyQ diseases [33]. As reported by Peng and collaborators, efforts to include patients from asymptomatic and preclinical stages should persist in order to obtain more accurate cut-off points in these stages of the illness [34]. Such efforts could help in giving a globally valuable preliminary assessment.

**Table 1 ijms-25-08074-t001:** Fluid biomarkers in SCA3 with stage-discriminative features.

Biomarker	Fluid	Controls	Preataxic/ Preclinical SCA3 Patients	Ataxic Patients	Associated Variables	Reference
PolyQ-ATXN3	Plasma	0.00 [0.00; 0.28] pg/µL (n = 34)	0.84 [0.48; 1.68] pg/µL (n = 4)	1.28 [0.03; 3.44] pg/µL (n = 41)	Perfect discrimination capacity between ataxic vs. controls. No clinical associations. No differences observed between presymptomatic and symptomatic.	[31]
		0.14 [0.1; 0.4] pg/mL (n = 15)	53.80 [40.28; 63.37] pg/mL (n = 11)	83.30 [55.38; 106.6] pg/mL (n = 45)	Positively correlated with SARA and negatively correlated with age of ataxia onset.	[35]
	CSF	0.00 [0.00; 0.04] pg/µL (median, n = 33)	0.04 [0.04; 0.07] pg/µL (median, n = 4)	0.13 [0.04; 0.46] pg/µL (median, n = 45)	Perfect discrimination between SCA3 patients and controls. No associations with other variables found. The median concentration was higher in symptomatic vs. asymptomatic.	[31]
		0.11 [0.1; 0.4] pg/mL (n = 18)	Not reported. Preataxic levels were not different from ataxic (n = 5).	5.48 [4.85; 6.67] pg/mL (n = 12)	CSF perfectly discriminated between controls and ataxic carriers.	[35]
	Urine	Not reported	Not reported.	Not reported. PolyQ-ATXN3 is higher in symptomatic patients than in those with other types of ataxia and controls.	Mild correlation with plasma levels.	[36]
NfL	Serum	Cohort B: 6.88 ± 2.72 pg/mL (n = 91)	15.03 ± 7.49 pg/mL (n = 26)	37.56 ± 13.47 pg/mL (n = 90)	Positively related to disease severity. Positively associated with SARA and ICARS. Negatively associated with cerebellar and brainstem volumes.	[37]
		Cohort 1 (ESMI): 8.6 [5.7;11.7] pg/mL (n = 77) Cohort 2 (EuroSCA/RiSCA):19.4 [15.1:25.4] pg/mL (n = 48)	29.1 [15.9:43.7] pg/mL (n = 8) 47.3 [25.5:78.0] pg/mL (n = 14)	34.8 [28.3:47.0] pg/mL (n = 75) 85.5 [70.2:100.2] pg/mL (n = 27)	Positive correlation with age, CAG repeat length and longitudinal SARA score. Prediction of time to onset. Preconversion stage delineation. Differentiation between early and late preataxic stages.	[38]
		8.24 [5.92–10.84] pg/mL (n = 185)	21.84 [18.37–23.45] pg/mL (n = 20)	36.06 [30.04–45.90] pg/mL (n = 198)	Negative correlation with gray matter in left precentral gyrus and paracentral lobule, as well as mean diffusivity in widespread matter tracts.	[34]
		7.43 pg/mL (n = 14)	--	35.33 pg/mL (n = 20)	Positively correlated with SARA and ICARS, speech disorders and limbic kinetic function.	[39]
		10.24 ± 4.48 pg/mL (n = 19)	Not reported.	34.92 ± 10.95 pg/mL(n = 20)	Optimal cut-off: 16.04 pg/mL. Baseline levels correlate with disease duration and SARA. Concentrations persisted after a 2-year follow-up.	[40]
	Plasma	11.20 [3.31: 33.56] pg/mL (n = 30)	15.42 [11.05: 28.68] pg/mL (n = 4)	30.1 [17.24: 69.9] pg/mL (n = 41)	Discrimination between SCA3 patients and controls.	[31]
		2.31 [0.83] log pg/mL (n = 172)	2.70 [0.47] log pg/mL (n = 23)	3.26 [0.46] log pg/mL (n = 120)	Predictors: age and number of CAG repeats account for 4.2% variability in ataxic and 30.63% in preataxic.	[41]
		5.7 [4.3; 7.2] pg/mL (n = 39)	19.8 [13.9; 27.3] pg/mL (n = 24)	31.4 [26.4; 36.4] pg/mL (n = 64)	Correlation with disease duration, SARA, INAS and CCFS, as well as diplopia.	[23]
	CSF	Cohort A 471.70 ± 210.40 pg/mL (n = 17)	Not reported.	4262.00 ± 1762.00 pg/mL (n = 9)	Higher in manifest SCA3 patients. CSF was 102X higher than serum.	[37]
		449 [137: 1512] pg/mL (n = 34)	1352 [1019: 1398] pg/mL (n = 4)	3569 [1413: 6837] pg/mL (n = 46)	NfL discriminates symptomatic SCA3 patients from asymptomatic carriers and controls.	[31]

Abbreviations: NfL = neurofilament light chain; CSF = cerebrospinal fluid; ICARS = International Cooperative Ataxia Rating Scale; SARA = Scale for the Assessment and Rating of Ataxia; INAS = Inventory for Non-ataxia Signs; CCFS = Composite Cerebellar Functional Severity Score.

PolyQ-ATXN3 was the second most frequently measured biomarker in SCA3. However, there are few studies that have evaluated their concentrations in plasma (n = 2), urine (n = 1) and CSF (n = 2). This marker has been successfully used to discriminate between controls and ataxic patients, but its discrimination capacity between the preclinical stage and ataxic and controls remains unclear. Only one study showed differences between the preclinical and ataxic stages in plasma [35]. In addition, the correlation with clinical variables like SARA or disease onset is not consistent. PolyQ-ATXN3 can be a valuable biomarker for evaluating the efficacy of novel, targeted therapies against expanded *ATXN3*, such as siRNA [31], miRNA [42] and clustered regularly interspaced short palindromic repeats (CRISPR)-based technology [43,44]. PolyQ-ATXN3 has also been successfully measured in peripheral blood mononuclear cells (PBMCs) [45] and is therefore a potential trait and state marker for tracking disease progression and the efficacy of therapeutic strategies.

Some patients with SCA3 show comorbid manifestations related to other movement disorders [46,47,48,49,50]. This is why studying different biomarker profiles could be a more specific approach for better differential diagnoses. For example, NfL is not specific to SCA3 [51,52,53,54,55,56,57]. Neuron-specific enolase (NSE) and S100B are markers of progressive cell damage [58], but only the former correlates with SARA and ICARS. Therefore, by themselves, some biomarkers like NfL, phosphorylated heavy chain neurofilament (pNfH) and NSE may not be useful for differentiating SCA3 from other movement disorders; instead, their clinical use could rely on identifying associations with other biomarkers and building a profile that must be carefully interpreted and correlated with clinical findings and imaging features.

There are other biomarkers that have been measured in ataxic and control patients (Appendix A), some of which have shown relevant correlations with clinical variables that could render them functional value state biomarkers. For example, plasma t-tau has been correlated with CAG repeats and INAS. Furthermore, through the SCA3 stages, the t-tau and p-tau^181^ concentration trends in CSF seem to follow an inverted U-shape curve, suggesting that they could be predictive biomarkers before the symptomatic stage. Moreover, in patients with SCA3, β-amyloid protein at amino acid 42 (Aβ_42_) is different from other neurological disorders, indicating that this may be useful for differentiating between SCA3 and cognitive and movement disorders. Perhaps they share common pathophysiological pathways that could be included in clinical trials.

We also found other biomarkers involved in the pathogenic pathways common to other neurodegenerative disorders, such as oxidative stress and metabolism disorders (Appendix A). While some of these potential state biomarkers can provide information about the disease’s pathomechanism, there are two limitations. The first is that most of them are only measured in controls and ataxic patients, reducing the information required for understanding the preclinical stage. The second is their lack of correlation with clinical features. This is the case for Ubiquitin Carboxy-terminal Hydrolase L1 (UCHL1), pNfH, ROS, superoxide dismutase (SOD), catalase, glutathione peroxidase (GSH-Px), eotaxin and some metabolic markers. Therefore, these biomarkers may require more research before being considered for stage discrimination. Other biomarkers like Glial Fibrillary Acidic Protein (GFAP) and the carboxyl terminus of Hsp-70 protein (CHIP) have clinical correlations that make them worth considering in evaluating state biomarker profiles.

Figure 3a shows the trajectories of the levels of different plasma and serum biomarkers in the different SCA3 stages, demonstrating correlations with the clinical variables shown in Table 1 and Appendix A. Interestingly, these biomarkers follow different patterns during the asymptomatic, preclinical and ataxic stages. Some of them, like NfL, polyQ-ATXN3, GFAP and t-tau, increase until patients reach the ataxic stage. However, some may decrease thereafter, as is the case for t-tau in plasma in symptomatic patients older than 40 years. Biomarker concentrations can also vary in CSF. The polyQ-ATXN3 and NfL levels in CSF followed a similar pattern to those in plasma (Figure 3b). The levels of t-tau and p-tau^181^ were higher in preclinical patients than in controls, but decreased in the ataxic stage. Unfortunately, only a few biomarkers have been evaluated in the asymptomatic stage, reducing the possibility of identifying putative trait biomarkers and drawing conclusions about the patterns that biomarker levels follow throughout SCA3 progression. Nonetheless, comparisons with controls and between clinical and symptomatic stages can illuminate the state biomarker end point, if not the whole process.

Interestingly, novel markers like exosomal miRNAs have been differentially found in plasma and CSF in patients with SCA3. In particular, mir-7014 could be a relevant trait and state biomarker, provided it is evaluated in asymptomatic and preclinical patients [59]. Recent studies, particularly those using state-of-the-art transcriptomics [60], have identified some genes, like splicing factor SWAP (SFSWAP), scaffold attachment factor B2 (SAFB2) and latent transforming growth factor beta-binding protein 4 (LTBP4), with the potential to differentiate between the preclinical, the ataxic and possibly the asymptomatic stages. It is also important to include an assessment of these biomarkers, which could have a dual function in terms of both trait and state, in clinical trials. Some clinical trials have been conducted (Appendix A), but they lack assessments of fluid biomarkers. The complex nature of SCA3 and its progression demands a successful therapeutic approach that must be delivered in the preclinical stage. Considering the psychological burden of testing positive for such a neurodegenerative illness, diagnosing SCA3 presymptomatically would be crucial in preserving neurological functions [61,62].

One of the main limitations in identifying a “gold-standard” trait, state or pharmacodynamic biomarker is the absence of measurements at the preclinical stage. This could be due to the fact that SCA3 has a low prevalence, thus reducing the number of patients without ataxic symptoms attending medical appointments; this may be why asymptomatic and preclinical cohorts are underdiagnosed. Therefore, it is worth increasing efforts to achieve wider recruitment and a higher statistical power. Nonetheless, examining the current cohorts is still a valuable approach to assessing molecular, imaging and clinical biomarkers. For instance, patients can show white-matter motor network alterations prior to the ataxic stage [63], but there has been insufficient study of their association with molecular biomarkers. It is also important to consider that SCA3 progression and staging will be influenced by age, sex, non-ataxic symptoms (cognitive alterations, depression, pain, autonomic dysfunction, etc.), individual variability and environment, and future studies should include this information when reporting data.

Finally, there are differences in certain technical aspects, like the antibodies and immunoassay platforms used to detect and quantify these biomarkers. These narrow the options for performing a meta-analysis or establishing cut-off point statistics. Future investigations should consider new methodological and statistical approaches, as well as reporting data using an international, common nomenclature, in order to circumvent these limitations. This could be achieved via collaboration between multiple research centers, including global population cohorts. In addition, ways of classifying patients when reporting data are inconsistent; some studies use the terms “asymptomatic” vs. “symptomatic”, preataxic vs. ataxic, manifest vs. non-manifest or categorizations based on SARA scores.

## 4. Conclusions

This study has shown the different fluid biomarkers that can be included in the assessment profile for SCA3 diagnosis and prognosis. This could be useful for thoroughly evaluating disease staging and therapeutic efficacy. NfL and polyQ-ATXN3 provide a closer view of disease progression, and they have been assessed as state biomarkers. This being said, more information about their patterns during the preclinical stages and ataxic onset is needed in order to clarify their use as trait biomarkers. These may be complemented by other biomarkers that have been successfully evaluated in the preclinical stage, such as the novel genes found in transcriptomics. Moreover, the sub-classification of the asymptomatic and preataxic stages should also include clinical evaluations and imaging correlations. Assessing fluid biomarkers could be a good start in validating SCA3 diagnosis and prognosis, but they may not be completely conclusive when considered alone. Furthermore, increased preataxic and asymptomatic group sizes should be considered in future investigations, and diagnostic procedures must be improved in order to detect patients in these stages. In future endeavors in developing clinical management guidelines and finding therapeutic targets, evaluating trait and state biomarkers could be a first step. Being armed with a battery of sensible and specific molecular biomarkers may open up more options for identifying correlations between clinical and neuroimaging variables. Such approaches could potentially help to understand SCA3 pathological pathways and find appropriate time windows for therapeutic interventions.

## Figures and Tables

**Figure 1 ijms-25-08074-f001:**
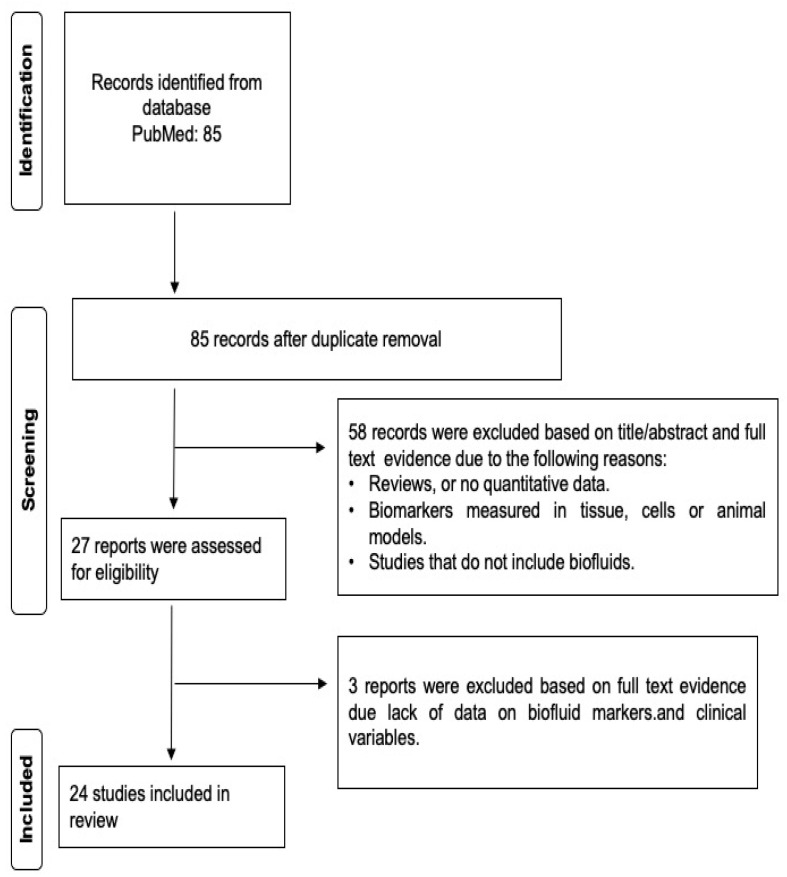
Flow diagram of EPPI v6 study selection following PRISM and STROBE guidelines.

**Figure 2 ijms-25-08074-f002:**
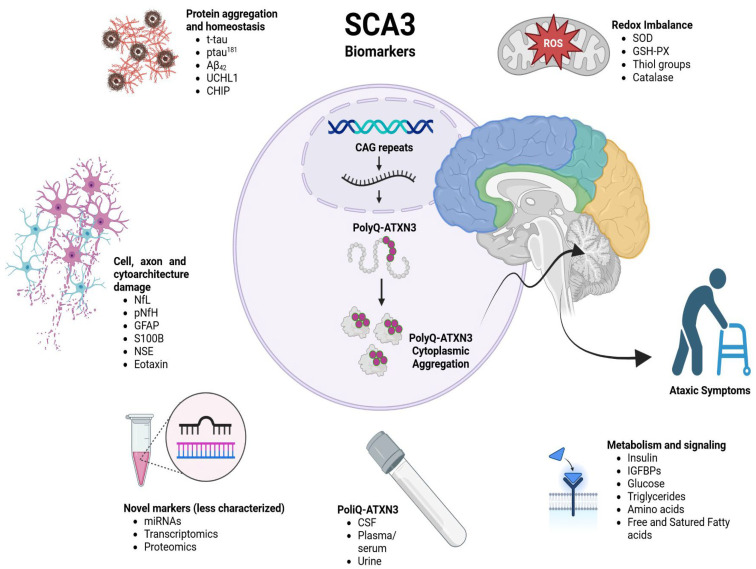
SCA 3 and biofluid markers. NfL = neurofilament light chain; CSF = cerebrospinal fluid; Aβ_42_ = β-amyloid protein at amino acid 42; GFAP = glial fibrillary acidic protein; UCHL1 = ubiquitin carboxy-terminal hydrolase L1; pNfH = phosphorylated neurofilament heavy; IGF-1 = insulin grown factor 1; IGFBPs = IGF-binding proteins; NSE = neuron-specific enolase; CHIP = carboxyl terminus of Hsp-70 protein; SOD = superoxide dismutase; and GSH-Px = glutathione peroxidase.

**Figure 3 ijms-25-08074-f003:**
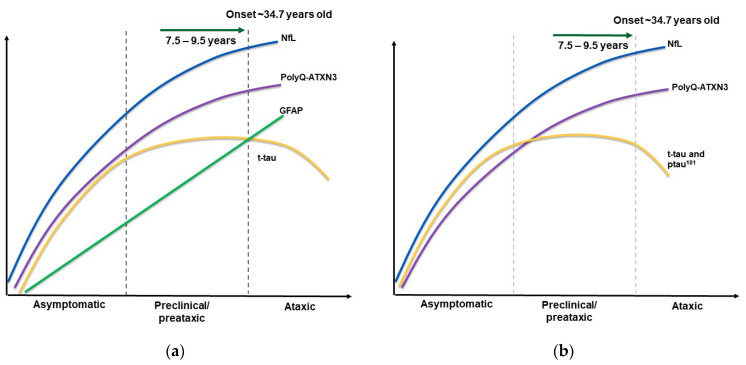
SCA3 stages are indicated by gray dashed lines. The Y axis represents the levels of biomarkers, and the X axis depicts disease progression (**a**). CSF biomarkers in SCA3 stages. Disease progression is shown along the X axis, and biomarker levels along the Y axis. Stages are indicated by gray dashed lines (**b**).

## Data Availability

No new data were created or analyzed in this study. Data sharing does not apply to this article.

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
