# Peer review of "Specific Biomarkers in Spinocerebellar Ataxia Type 3: A Systematic Review of Their Potential Uses in Disease Staging and Treatment Assessment"

_ijms, 2024, doi:10.3390/ijms25158074_

Round 1
Reviewer 1 Report
Comments and Suggestions for Authors
The review article by Dr Soto-Piña et al., entitled “Specific biomarkers in SCA3: systematic review of potential uses in disease staging and treatment assessment”, addresses the current literature concerning the advances in spinocerebellar ataxia type 3 (SCA3) fluid biomarkers. This interest comes from the need of reliable biomarkers for SCA3 to be used to evaluate disease progression and efficacy of therapeutic approaches applied in worldwide clinical trials.
The Authors found that neurofilament light chain (NfL) is a potential biomarker candidate to monitor disease stage progression, polyQ-ATXN3 (ataxin 3) may be used as a pharmacodynamic marker for therapies targeted to polyQ-expanded ATXN3, while generic markers of neurodegeneration and oxidative stress may be monitored as indices of SCA3 pathomechanisms.
The review is accurate and well written, and it covers appropriately the main issue. I have only minor changes to suggest to the Authors:
-The results of this comprehensive review were obtained by analyzing data related to 1069 ataxic patients, 142 pre-ataxic patients and 1078 healthy controls. The number of pre-ataxic patients is far lower than other groups. To enlarge the size of preataxic group could be help to reveal some molecules that are key markers for pre-clinical diagnosis and/or disease progression.
-The organization of the review is not properly carried out. I think that sections as Results and Materials are not required in a review article. Anyway, I suggest that sections that are within the Discussion and contain references to Figures and Tables are moved in the Results, and Discussion is modified according to these changes.
Author Response
We greatly appreciate the comments from the reviewer, they were very helpful to improve the content of this review as well as the relevance in the field. We addressed the comments from the reviewer as follows:
Comment 1. The results of this comprehensive review were obtained by analyzing data related to 1069 ataxic patients, 142 pre-ataxic patients and 1078 healthy controls. The number of pre-ataxic patients is far lower than other groups. To enlarge the size of preataxic group could be help to reveal some molecules that are key markers for pre-clinical diagnosis and/or disease progression.
Response 1. We have approached this meaningful criticism of the reviewer. In the revised version of our manuscript, the number of patients increased, because some more recent publications were added in this study (Supplementary table S1). The sample size of participants is now included in Materials and Methods section (see lines 127-129). However, in agreement to reviewer, the size of the preataxic group is usually low due the low prevalence of SCA3 (lines 37-38) and because patients in general do not attend to medical appointment unless the symptoms have already appeared. We thank the Reviewer for this opportune observation, and we have now stated this important point in Discussion section (lines 239-244).
Comment 2. The organization of the review is not properly carried out. I think that sections as Results and Materials are not required in a review article. Anyway, I suggest that sections that are within the Discussion and contain references to Figures and Tables are moved in the Results, and Discussion is modified according to these changes.
Response 2. In line with this opportune suggestion, we now present a new organization of the current version showing a simple and more logical sequence for the reader. The organization of the review was modified according to the guidelines suggested by the editor: introduction, methods, results/discussion and conclusion. The manuscript was also English and layout edited by an expert from MDPI author services.

Reviewer 2 Report
Comments and Suggestions for Authors
Abstract:
ü The abstract tries to cover too many aspects, making it difficult to identify the primary focus. It begins with a broad overview of SCA3 and then abruptly shifts to discussing biomarkers without a smooth transition.
ü Including detailed technical terms such as "C-terminal coding region (exon 10) of ataxin-3 (ATXN3) gene" is unnecessary and overcomplicates the abstract.
ü Information about ATXN3 protein aggregation and neuropathological changes is repeated and could be consolidated for clarity.
ü Phrases like "some have been hindered by the lack of biomarkers" and "sustained capacity as a pharmacodynamic marker" are vague and need more precise language.
ü The abstract mentions the importance of neurofilament light chain (NfL) and polyQ-ATXN3 as biomarkers without adequately explaining their significance or how they were identified.
ü The study's aims are not clearly defined; it’s unclear whether the focus is on identifying new biomarkers, validating existing ones, or another objective.
ü The writing style is inconsistent, with some sentences being too technical and others too vague.
Introduction:
ü The introduction lacks a clear structure, transitioning abruptly between topics, making it hard to follow.
ü It provides excessive technical details early on, which might not be necessary for setting up the study's context.
ü Information about the disease's characteristics and neuropathological features is repeated.
ü While it provides extensive background information on SCA3's clinical and pathological features, it lacks focus on how these details are relevant to biomarker study.
ü Although it mentions the limitations of current treatments and monitoring methods, it does not clearly define the specific gap this study aims to address.
ü The study's purpose is not clearly outlined until later in the introduction, potentially confusing readers about the study’s aims.
ü Numerous studies are referenced without smooth integration, making the introduction feel like a list of facts rather than a cohesive narrative.
ü The terminology and writing style vary, disrupting the reader's understanding. For instance, “polyQ-ATXN3” is introduced without clear definition.
Results:
ü The results mention that the I2 index in plasma was 0 in preclinical and ataxic patients, indicating no variability across studies, but the serum data showed high variability (I2 > 96%), suggesting inconsistency in serum measurements due to differences in study design, sample handling, or analytical methods. High variability undermines the reliability of the serum NfL data.
ü No CSF data for preclinical patients limits comprehensive conclusions about NfL levels across all disease stages, making it difficult to understand the biomarker's behavior in early SCA3 stages.
ü The results highlight the lack of longitudinal studies showing NfL dynamics over time. Longitudinal data is crucial for understanding NfL level changes and their correlation with disease progression. Cross-sectional data, while useful, cannot substitute for long-term tracking insights.
ü The study does not account for potential confounding factors like age, sex, and non-ataxic symptoms (e.g., cognitive alterations, depression, autonomic dysfunction). These factors can influence NfL levels and should be controlled for in the analysis to ensure observed differences are due to SCA3.
ü NfL is not specific to SCA3 and shows alterations in various neurodegenerative disorders, which could lead to misinterpretation of elevated NfL levels if used alone as a diagnostic or staging tool. NfL levels need to be interpreted in the context of a broader biomarker profile and clinical presentation.
ü Differences in technical aspects such as antibodies and immunoassay platforms used to detect and quantify NfL can lead to variability in results and complicate direct comparisons across studies. Standardization of methods and protocols is needed to improve biomarker measurement reliability.
ü Small sample sizes in some studies could contribute to variability and limit the generalizability of the findings. Larger, multicenter studies are needed to validate the results and ensure they are representative of the broader SCA3 population.
ü The absence of data at the preclinical stage hinders the ability to set accurate cut-off points for different SCA3 stages. Establishing these cut-off points is essential for early diagnosis and monitoring disease progression, but current data limitations prevent this.
Materials and Methods:
1. Literature Search Strategy:
ü The search terms "Spinocerebellar ataxia type 3" and "Machado-Joseph Disease" combined with "biomarkers" might exclude relevant studies using different terminology or focusing on related aspects of the disease. A broader, more inclusive search strategy could yield a more comprehensive dataset.
ü Limiting inclusion to studies published in English introduces language bias and excludes relevant research in other languages.
ü The search cut-off at September 30, 2023, means recent advancements or studies published afterward are not considered, potentially missing the latest findings.
ü While EPPI Reviewer is a useful tool for records discrimination, the criteria and methods used within the tool need to be clarified for transparency and reproducibility.
2. Data Extraction and Management:
ü Concentrations from the asymptomatic stage were not included in the main tables due to insufficient data, which could skew the understanding of biomarker progression in early disease stages.
ü The broad definition of preclinical stages includes various symptoms that might not be specific to SCA3, leading to potential misclassification and inconsistent data grouping.
ü The omission of stage classification in Table 2 due to non-uniform biomarker measurements across studies indicates a significant gap, hindering comprehensive analysis and comparisons.
3. Statistical Analysis for NfL Levels:
ü Although the original studies considered confounders like sex and age, not adjusting for these factors in the meta-analysis could introduce bias. Meta-analyses should independently control for these variables to ensure accurate results.
ü Reliance on the I2 statistic to measure between-study heterogeneity is standard practice, but high heterogeneity (as indicated in serum NfL) suggests variability in study methods and populations that could undermine the validity of pooled estimates.
ü Excluding two studies due to lack of independent cohorts might limit the dataset. A sensitivity analysis including these studies could provide insights into their impact on the overall results.
Conclusion:
ü While the study acknowledges that biomarkers like NfL and polyQ-ATXN3 cannot fully explain the disease's complexity, it emphasizes their potential without adequately addressing their limitations. The conclusion should balance the potential utility of biomarkers with the need for comprehensive diagnostic and monitoring approaches.
ü The recommendation for sub-classifying the preataxic stage using fluid biomarkers, SARA scores, and imaging approaches is logical but requires more robust data. Current inconsistencies in biomarker measurement across studies weaken this recommendation.
ü Developing guidelines for patient prognosis management and therapeutic interventions based on fluid biomarkers is premature given current data gaps and variability. More extensive and standardized research is needed before establishing such guidelines.
ü Determining more opportune time windows for therapeutic interventions is valuable, but depends on consistent and reliable biomarkers. The existing variability in biomarker data weakens the foundation for making such determinations.
Comments on the Quality of English LanguageExtensive editing of English language required
Author Response
We would first like to thank the Reviewer for their helpful comments and suggestions regarding the manuscript.
Abstract:
Comment 1: The abstract tries to cover too many aspects, making it difficult to identify the primary focus. It begins with a broad overview of SCA3 and then abruptly shifts to discussing biomarkers without a smooth transition.
Response 1: We are in accord with this concern from the reviewer. We modified the abstract in order to improve the narrative and to covey the relevance of this review.
Comment 2: Including detailed technical terms such as "C-terminal coding region (exon 10) of ataxin-3 (ATXN3) gene" is unnecessary and overcomplicates the abstract.
Response 2: In accordance with this suggestion, this term was removed (Line 20). The narrative of the abstract was modified to reduce technical terms.
Comment 3: Information about ATXN3 protein aggregation and neuropathological changes is repeated and could be consolidated for clarity.
Response 3: This information was clarified Lines 20-21.
Comment 4: Phrases like "some have been hindered by the lack of biomarkers" and "sustained capacity as a pharmacodynamic marker" are vague and need more precise language.
Response 4: Narrative was changed in the abstract to clarify those phrases, for example lines 23-31.
Comment 5: The abstract mentions the importance of neurofilament light chain (NfL) and polyQ-ATXN3 as biomarkers without adequately explaining their significance or how they were identified.
Response 5: According with this pertinent observation, results section in the abstract was modified accordingly to the main findings regarding NfL meta-analysis (lines 25-29).
Comments 6: The study's aims are not clearly defined; it’s unclear whether the focus is on identifying new biomarkers, validating existing ones, or another objective.
Response 6: The aim was clarified lines 24-25.
Comment 7: The writing style is inconsistent, with some sentences being too technical and others too vague.
Response 7: In concordance with this opportune observation, the abstract was re-written and revised by an English editing expert from MDPI author services.
Introduction:
Comment 1: The introduction lacks a clear structure, transitioning abruptly between topics, making it hard to follow.
Response 1: In line with this opportune suggestion, the introduction was modified accordingly to reviewer suggestions and adjusted to the updates in the methods, results and conclusion.
Comment 2: It provides excessive technical details early on, which might not be necessary for setting up the study's context.
Response 2: The technical terms about molecular and clinical features were removed from the introduction, for example lines 39-50.
Comment 3: Information about the disease's characteristics and neuropathological features is repeated.
Response 3: This information was reduced in general throughout the introduction.
Comment 4: While it provides extensive background information on SCA3's clinical and pathological features, it lacks focus on how these details are relevant to biomarker study.
Response 4: We are totally in accord with this meaningful criticism. A brief description about the relevance was carried out in lines 44-50, 53-56, 59-63. The relevance to study fluid biomarkers relies on the need to follow the progression of the disease, to identify therapeutic targets and to assess therapeutic efficacy in clinical trials. It is required to identify trait and state biomarkers within the variety of biomarkers in the field.
Comment 5: Although it mentions the limitations of current treatments and monitoring methods, it does not clearly define the specific gap this study aims to address.
Response 5: We define some gaps in lines 44-56, 59-64.
Comment 6: The study's purpose is not clearly outlined until later in the introduction, potentially confusing readers about the study’s aims2
Response 6: Effectively, we failed to explain the study’s purpose. The revised version of the manuscript now includes the aim of the study in lines 64-66 of the introduction section. It was included at the end because we explained in the 2nd and 3rdparagraphs of the introduction the two main gaps in the use of biomarkers: disease staging and therapeutic efficacy monitoring.
Comment 7: Numerous studies are referenced without smooth integration, making the introduction feel like a list of facts rather than a cohesive narrative.
Response 7: The references in the introduction were modified accordingly to the updated narrative.
Comment 8: The terminology and writing style vary, disrupting the reader's understanding. For instance, “polyQ-ATXN3” is introduced without clear definition.
Response 8: PolyQ-ATXN3 definition was clarified (lines 39-41). Homogeneity in the narrative was improved and the manuscript was submitted for English and layout editing by an expert from MDPI author services.
Results:
Comment 1: The results mention that the I2 index in plasma was 0 in preclinical and ataxic patients, indicating no variability across studies, but the serum data showed high variability (I2 > 96%), suggesting inconsistency in serum measurements due to differences in study design, sample handling, or analytical methods. High variability undermines the reliability of the serum NfL data.
Response 1: We agree with this comment. I2 results for plasma, serum and CSF are different, this could indicate that NfL assessment may be more reliable when using plasma or CSF than serum; and when making specific comparisons, for example preclinical vs ataxic and control vs ataxic (lines 132-144). According with this pertinent observation, a detailed explanation was added.
Comment 2: No CSF data for preclinical patients limits comprehensive conclusions about NfL levels across all disease stages, making it difficult to understand the biomarker's behavior in early SCA3 stages.
Response 2: While it is a limitation not to have multiple studies to perform a meta-analysis about NfL in CSF, there is still one study (Prudencio, et al. 2020, as seen in table1) that shows data about CSF NfL levels in the preataxic stage. We added a brief discussion that considers these aspects. The results indicate NfL levels are higher in the symptomatic stage than the preclinical, more importantly ROC curves analysis shows that NfL has discriminative capacity between ataxic vs presymptomatic and controls. We added this information in lines 146-148. In addition, the procedure to obtain CSF is invasive and in many cases patients decline to undergo lumbar punction for this purpose, as in NfL (lines 139-140).
Comment 3: The results highlight the lack of longitudinal studies showing NfL dynamics over time. Longitudinal data is crucial for understanding NfL level changes and their correlation with disease progression. Cross-sectional data, while useful, cannot substitute for long-term tracking insights.
Response 3: In concordance with this opportune observation, we mentioned that NfL dynamics through disease progression have not been evaluated as in other polyQ diseases (see lines 150-153).
Comment 4: The study does not account for potential confounding factors like age, sex, and non-ataxic symptoms (e.g., cognitive alterations, depression, autonomic dysfunction). These factors can influence NfL levels and should be controlled for in the analysis to ensure observed differences are due to SCA3.
Response 4: We agree with this comment. Unfortunately, most of the studies do not present specific data about those cofounding variables, there is no homogeneity in the way the results are reported, in fact this could be a limitation for all the biomarkers (lines 248-251). A suggestion for future studies about biomarkers may be to include this kind of information in the reports. Our results about NfL are still valid since we controlled by methodology (Lines 101-102), under the classification of SCA3 stages and an objective statistical analysis for heterogeneity (lines 104-114).
Comment 5: NfL is not specific to SCA3 and shows alterations in various neurodegenerative disorders, which could lead to misinterpretation of elevated NfL levels if used alone as a diagnostic or staging tool. NfL levels need to be interpreted in the context of a broader biomarker profile and clinical presentation.
Response 5: Following this meaningful observation, we now include a brief discussion. NfL levels is a marker which is analyzed in similar neurological pathologies, some related to polyQ. This is why diagnostics or prognosis of patients with SCA3 should not solely rely on NfL, but they should include other molecular and imaging biomarkers as well as a thorough clinical assessment (for example Lines 174-183). More importantly NfL is the biomarker measured in fluids with more clinical associations (Table 1, lines 148-150).
Comment 6: Differences in technical aspects such as antibodies and immunoassay platforms used to detect and quantify NfL can lead to variability in results and complicate direct comparisons across studies. Standardization of methods and protocols is needed to improve biomarker measurement reliability.
Response 6: We thank the reviewer for this this opportune observation. We now better describe these suggestions. In this particular meta-analysis we only included studies performed with the platform SIMOA (Single Molecule Array) to assess NfL (lines 101-102), which has been the gold standard to measure it. Therefore, our results of NfL account for the homogeneity in the methodology to assess NfL and the results rely on the variability of the population cohorts. Certainly, differences in methodologies may account for other biomarkers that were not included in our meta-analysis (252-258).
Comment 7: Small sample sizes in some studies could contribute to variability and limit the generalizability of the findings. Larger, multicenter studies are needed to validate the results and ensure they are representative of the broader SCA3 population.
Response 7: We are in accord with the reviewer, we now include a brief discussion. The small sample size is mainly because of the low prevalence of SCA3 (lines 37-38), it is a rare disease. However, there are already center collaboration groups around the world working on the elucidation of SCA3 biomarkers, including data of multiple cohorts (for example, García-Moreno et al., 2022, and Prudencio et al., 2020). Furthermore, the n in the different groups is relevant into the field of SCA3 (Lines 127-129, 239-246).
Comment 8: The absence of data at the preclinical stage hinders the ability to set accurate cut-off points for different SCA3 stages. Establishing these cut-off points is essential for early diagnosis and monitoring disease progression, but current data limitations prevent this.
Response 8: We agree with this comment. The small n in the preclinical stage of SCA3 could be mainly due to the low prevalence of SCA3 (lines 37-38), and this certainly can affect the analysis of cut-off points. Nonetheless, a preliminary analysis can be done even with small n, for instance Peng et al., 2020 presented cut-off points results with a n<20 for asymptomatic and preataxic stages. Lines 153-156, 240-246.
Materials and Methods:
Literature Search Strategy:
Comment 1: The search terms "Spinocerebellar ataxia type 3" and "Machado-Joseph Disease" combined with "biomarkers" might exclude relevant studies using different terminology or focusing on related aspects of the disease. A broader, more inclusive search strategy could yield a more comprehensive dataset.
Response 1: In accordance with this suggestion, the search strategy was updated (lines 71-73). In some cases, the use of some other terms like CSF, fluids, miRNA, reduces considerably the results, this is probably because SCA3 is a rare disease and the amount of studies is smaller than other diseases. This is also why in lines 73-75, the inclusion criteria of the studies are mentioned.
Comment 2: Limiting inclusion to studies published in English introduces language bias and excludes relevant research in other languages.
Response 2: Accordingly, the search was adjusted in Pubmed, there were no manuscripts written in other languages. Lines 73-76, 78-82.
Comment 3: The search cut-off at September 30, 2023, means recent advancements or studies published afterward are not considered, potentially missing the latest findings.
Response 3: Following this meaningful observation, Search strategy was updated now, and the cut-off changed to July 4th, 2024. Line 76 and Figure 1. Result section was also updated (Supplementary table S1, lines 184-206).
Comment 4: While EPPI Reviewer is a useful tool for records discrimination, the criteria and methods used within the tool need to be clarified for transparency and reproducibility
Response 4: EPPI Reviewer version 6 was used to perform records discrimination, the description on the process of selection in described in lines 76-82 and shown in figure 1. The diagram was built based on PRISM guidelines because EPPIv6 does not count with the option to create it.
Data Extraction and Management:
Comment 1: Concentrations from the asymptomatic stage were not included in the main tables due to insufficient data, which could skew the understanding of biomarker progression in early disease stages.
Response 1: According to the correct comment of the reviewer, this sentence was clarified and applies specifically for NfL (lines 94-99). Concentrations for NfL in asymptomatic or presymptomatic patients (from Prudencio et al., 2021 and Peng et al., 2020) are lower than preclinical and ataxic patients. This evidence does not interfere with reported biomarker progression.
Comment 2: The broad definition of preclinical stages includes various symptoms that might not be specific to SCA3, leading to potential misclassification and inconsistent data grouping.
Response 2: We agree with this comment, but the diagnosis of SCA3 usually includes the measure of the CAG repeat in ATXN3 which confirms a patient is a carrier of the mutation, so that renders specificity in the SCA3 diagnosis. Peng and collaborators (2020) showed a clear stage discrimination between the asymptomatic and clinical phases for NfL in SCA3 carriers (lines 153-156). While this could be a limitation, we included a column with clinical variables that show significant associations with the respective biomarker in Table 1 and Supplementary table S1.
Comment 3: The omission of stage classification in Table 2 due to non-uniform biomarker measurements across studies indicates a significant gap, hindering comprehensive analysis and comparisons.
Response 3: That is a limitation in the SCA3 studies because many biomarkers have only been measured in controls and ataxic patients. PolyQ-ATXN3 and NfL are the main biomarkers that have been measured in all the stages and with more similar trend patterns, which could indicate reproducibility in the studies. This is why they could be used to monitor disease progression and treatment efficacy. We modified the structure of the methods and results to clarify this issue (Supplementary table S1 and lines 97-99, 194-206, 216-233).
Statistical Analysis for NfL Levels:
Comment 1: Although the original studies considered confounders like sex and age, not adjusting for these factors in the meta-analysis could introduce bias. Meta-analyses should independently control for these variables to ensure accurate results.
Response 1: Unfortunately, the results in the manuscripts have not presented data by sex or age, or specific demographic information about them. Therefore, there is not sufficient information to perform that kind of testing in this meta-analysis. We attached some sentences that discuss these points. This could be a suggestion to consider in future studies (Lines 248-251). Regarding our NfL analysis, it was performed based on one methodology (SIMOA), and this is why the heterogeneity analysis was carried out, to have a general statistical description about the NfL studies and SCA3 stages (lines 101-105).
Comment 2: Reliance on the I2 statistic to measure between-study heterogeneity is standard practice, but high heterogeneity (as indicated in serum NfL) suggests variability in study methods and populations that could undermine the validity of pooled estimates.
Response 2: While the I2 statistic is considered as a descriptive statistic it is more powerful than other tests of heterogeneity (Lin, 2019). In this case since the number of studies is small, there could be an overestimation in I2 (12%) (T von Hippel, 2015). In our meta-analysis report some results showed high I2 (>90%) therefore, the real I2 could be close to 80% indicating. Therefore, the I2 is still high for the NFL results in some of the biofluids; but in others it is medium or low, indicating that those are still reliable for interpretation. Moreover, in our study is accompanied by mean, difference of means and p value (Supplementary tables S2 and S3). Lines 104-114, 130-140.
Comment 3: Excluding two studies due to lack of independent cohorts might limit the dataset. A sensitivity analysis including these studies could provide insights into their impact on the overall results.
Response 3: Those cohorts were already included in the analysis as the main cohorts from the publications included. However, those cohorts functioned as validation cohorts in the respective studies (Prudencio et al., 2020 and García Moreno et al., 2022). They were included as main cohorts from their respective publications. So, the patients were in fact included in the analysis. It was clarified in lines 102-104.
Conclusion:
Comment 1: While the study acknowledges that biomarkers like NfL and polyQ-ATXN3 cannot fully explain the disease's complexity, it emphasizes their potential without adequately addressing their limitations. The conclusion should balance the potential utility of biomarkers with the need for comprehensive diagnostic and monitoring approaches
Response 1: We totally agree with this meaningful criticism. We added a short sentence in “Discussion”. Our review shows biomarkers that have demonstrated to be useful to depict the ataxic stage and some even the preclinical stage, an assessment of biomarkers profile could be helpful to diagnose and follow up patients (see lines 265-270).
Comment 2: The recommendation for sub-classifying the preataxic stage using fluid biomarkers, SARA scores, and imaging approaches is logical but requires more robust data. Current inconsistencies in biomarker measurement across studies weaken this recommendation.
Response 2: The assessment in the preataxic stage is important to identify ways to prevent the disease. Current biomarkers are the closest approach to validate diagnosis and prognosis of patients. While it may be a good start, it is not fully conclusive as the reviewer comments, therefore we address this consideration (lines 270-273).
Comment 3: Developing guidelines for patient prognosis management and therapeutic interventions based on fluid biomarkers is premature given current data gaps and variability. More extensive and standardized research is needed before establishing such guidelines.
Response 3: We agree with this comment, we clarified this might be the beginning to develop future research and disease managing guidelines (lines 274-281). It is not currently conclusive to perform that.
Comment 4: Determining more opportune time windows for therapeutic interventions is valuable, but depends on consistent and reliable biomarkers. The existing variability in biomarker data weakens the foundation for making such determinations.
Response 4: We are in accord with the reviewer. The identification of objective biomarkers suggest they could have a potential use as state or therapeutic efficacy marker. However, it is necessary to integrate a biomarkers profile with different types (molecular and clinic) and this may help to achieve more specific clinical use. In this way, it might be possible to identify potential therapeutic targets and intervention windows in future studies. Lines 278-281.

Round 2
Reviewer 2 Report
Comments and Suggestions for Authors
Thanks for addressing the comments